# Synthesis and Conductivity Studies of Poly(Methyl Methacrylate) (PMMA) by Co-Polymerization and Blending with Polyaniline (PANi)

**DOI:** 10.3390/polym13121939

**Published:** 2021-06-11

**Authors:** Helyati Abu Hassan Shaari, Muhammad Mahyiddin Ramli, Mohd Nazim Mohtar, Norizah Abdul Rahman, Azizan Ahmad

**Affiliations:** 1Institute of Advanced Technology, Universiti Putra Malaysia, Serdang 43400, Selangor, Malaysia; helyati@uitm.edu.my (H.A.H.S.); a_norizah@upm.edu.my (N.A.R.); 2Faculty of Applied Sciences, Universiti Teknologi MARA Perlis Branch, Arau Campus, Arau 02600, Perlis, Malaysia; 3Geopolymer and Green Technology, Centre of Excellence (CEGeoGTech), Universiti Malaysia Perlis (UniMAP), Kangar 01000, Perlis, Malaysia; mmahyiddin@unimap.edu.my; 4Faculty of Engineering, Universiti Putra Malaysia, Serdang 43400, Selangor, Malaysia; 5Department of Chemistry, Faculty of Science, Universiti Putra Malaysia, Serdang 43400, Selangor, Malaysia; 6Department of Chemical Sciences, Faculty of Science and Technology, Universiti Kebangsaan Malaysia, Bangi 43600, Selangor, Malaysia; azizan@ukm.edu.my; 7Department of Physics, University of Airlangga, Surabaya 60115, Indonesia

**Keywords:** polyaniline, poly(methylmethacrylate), synthesis, conductivity

## Abstract

Poly(methyl methacrylate) (PMMA) is a lightweight insulating polymer that possesses good mechanical stability. On the other hand, polyaniline (PANi) is one of the most favorable conducting materials to be used, as it is easily synthesized, cost-effective, and has good conductivity. However, most organic solvents have restricted potential applications due to poor mechanical properties and dispersibility. Compared to PANi, PMMA has more outstanding physical and chemical properties, such as good dimensional stability and better molecular interactions between the monomers. To date, many research studies have focused on incorporating PANi into PMMA. In this review, the properties and suitability of PANi as a conducting material are briefly reviewed. The major parts of this paper reviewed different approaches to incorporating PANi into PMMA, as well as evaluating the modifications to improve its conductivity. Finally, the polymerization condition to prepare PMMA/PANi copolymer to improve its conductivity is also discussed.

## 1. Introduction

Poly(methyl methacrylate) is the synthetic polymer derived from methyl methacrylate (MMA) monomer, as depicted in Figure 1. Poly(methyl methacrylate), commonly abbreviated as PMMA, is a transparent polymer with good rigidity and dimensional stability. Since a German scientist named Otto Rohm first discovered its application in 1934 [1], this acrylic polymer has gained considerable interest as a substitute for glass due to these unique properties, such as good mechanical properties, light weight, shatter resistance, and ease of processing [2]. Moreover, PMMA is an insulator polymer, thus further widening its applications in different fields. However, throughout the years, there has been wide interest in fabricating conducting materials by using PMMA as the main material due to the demands to produce conducting material with balanced properties, like high conductivity and good mechanical properties [3]. More importantly, PMMA can be chemically co-polymerized or physically blended with other conducting polymers to produce a new material with synergetic properties between the two monomers [4].

Polyaniline (PANi) is one of the conducting polymers that is usually used in conjunction with PMMA. PANi consists of aniline monomers repeating units with a chemical formula of C_6_H_5_NH_2_ (Figure 2), and it can be chemically or electrochemically polymerized to produce a long chain of polyaniline.

PANi is becoming one of the most favorable polymers for the fabrication of conductive polymer due to the ease and low cost of synthesizing the polymer, as well as the environmentally friendly production and its chemical stability [5]. The synthesis of PANi is easy and simple to conduct, with no special equipment or precautions required [6]. Thus, polyaniline (PANi) has gained popularity as favorable polymers that have been used extensively as the main material for the fabrication of many conducting materials. PANi was first discovered in 1862 [7], when the bluish-black PANi powder was shown to dissolve in concentrated sulphuric acid but not in other more commonly used solvents such as water, alcohol, ether, and ammonia. Later, in 1910, PANi was further characterized into four different oxidation states by Arthur G. Green and his colleagues [8]. One of these states is the colorless leucomeraldine (LB), which is the fully reduced PANi, while the emeraldine base (EB) state with a violet-blue form is the half oxidized PANi, and the purple pernigraniline (PB) is the fully oxidized form of PANi. However, the only fully conducting form of PANi is the emeraldine salts (ES) state. The dark green ES is produced from the protonation of EB. Protonation is done by doping the PANi in an acidic medium with a pH of around 3 to 4, such as hydrochloric acid [9] and sulphuric acid [10].

Nevertheless, the conductivity of conjugated polymers is relatively low, due to the fact that the covalently bonded polymers have no valance band to serve as a pathway for electrons to move around. Usually, a doping process is performed to amplify the electron movements in an orbital. This is carried out by either the oxidation (p-doping) or the reduction (n-doping) process. The conductivity of PANi can be amplified by doping the polymer with various acids [11,12,13]. The acid contains charge carriers that provide empty spaces to be filled by the electrons. Thus, the flow of charges occurs, as the charges can be transported more easily and conduct electricity [14]. The effects of the doping level are more pronounced on the electrical conductivity than on the thermal conductivity of PANi, thereby greatly affecting the ratio that determines the thermoelectric efficiency [15].

Polyaniline, in its salt form, exhibits conductivity at around 15 S/cm [16]. Thus, throughout the years, PANi has been incorporated with PMMA for various applications, including electrochemistry and optoelectronics devices [17,18,19], sensors [20,21,22], and artificial muscles [23,24]. The incorporation methods include co-polymerization, blending, composite, and nano-composite. The mixture produced from the mixing of conducting PANi with PMMA-insulating phases enables the control of the electrical characteristics and increases the conducting phase above the percolation threshold to a few orders of magnitude [25]. In addition, the conductivity of the produced copolymer depends on several factors. The previous literature reported that the conductivity in such PMMA/PANi copolymers could be controlled in a wide range (more than 8–10 orders of magnitude) by altering the amount of PANi incorporated [26]. Hence, the conductivity increases systematically with increasing concentration of PANi until the optimum saturation level is reached [27]. The optimum amount of PANi content yields a low percolation threshold for the PMMA-PANi components due to the conducting network formation in the dielectric matrix inside the host polymer. In addition, the electrical conductivity increases due to the increasing polyaniline content, which subsequently increases the density and mobility of the charge carriers [28].

This review focuses on providing information on the different approaches used in fabricating PMMA/PANi copolymer, especially in the form of a thin film. Since the conductivity of PANi depends on many factors, such as the degree of oxidation, the protonation process, and also the number of electrons, this review also highlights the recent various modifications and improvements that can be performed during the synthesis process to enhance the conductivity of PMMA/PANi copolymer. Finally, this review also presents future prospects for the high conductivity material development through the novel use of PMMA/PANi copolymers.

## 2. Poly(Methyl Methacrylate)

### 2.1. Conducting PMMA

PMMA is chemically polymerized from the methyl methacrylate monomer. It is widely known as a good insulator due to its high resistivity (>2 × 10^15^ Ω·cm) [29]. Moreover, PMMA has added advantages, like having inherent mechanical stability and optical characteristics, which are important for the development and application as a conducting material [30]. Generally, the main focus is to produce new material that can conduct electrical currents efficiently but at the same time exhibit good mechanical and optical properties [31]. The bulk polymerization reaction initiated with benzoyl peroxide initiator (BPO) was usually employed, rather than the emulsion polymerization reaction [32], in order to fully exploit these outstanding properties of PMMA.

### 2.2. Mechanical Properties

The conductivity of conjugated polymers is due to the conjugated delocalized π electrons that develop the valence electron conduction bands throughout the backbone of the polymer [33]. Despite its outstanding optical and electrical features, the conjugated polymers face problems in their processing ability and performance [34]. Therefore, the incorporation of PMMA into the conjugated polymers is an effective way to improve the mechanical strength, as the PMMA contributes high mechanical stability to the combination [32]. Rozik et al. (2016) [35] reported a revolutionary solution-casting fabrication process that disperses different amounts of PANi into a solution containing the poly(methyl methacylate)/polystyrene (PMMA/PS) blends. The prepared blends contained the highest PMMA/PS loadings of 50%/50% weight ratio and gave high tensile strength in comparison with the other blends. However, the tensile strength value, as expected, decreases with the increasing PANi concentration due to the aggregation of PANi inside the polymer matrix. Similarly, 50% weight ratio of PMMA and 50% weight ratio of PS were incorporated into the PANi matrix to reinforce the structure, as such a ratio has proven to create polymer blends that exhibit good tensile strength and electrical properties [36]. The electrochemical oxidative method was employed by Francis et al. (2020) to produce PMMA/PANi blends through the use of hydrochloric acid as dopants [30]. The analysis using compression tests indicated that the mechanical strength increases considerably with the increase in PMMA ratio, with the compression strength of 0.08 N/m^2^ recorded for PANi, and the strength was improved to 0.14 N/m^2^ for PANi with 20% PMMA by weight.

### 2.3. Dispersibility

PANi is the simplest and cheapest to be synthesized, as compared to the other types of conjugated conducting polymers [37]. Despite all of these advantages, its application is limited because of its poor dispersibility of PANi caused by its rigid backbone in most of the organic solvents [38]. As a solution to this, the PANi structure can be modified with other polymers, such as PMMA, by using a radical polymerization reaction [39]. The reacting mixture of PMMA/PANi was reported to form a stable dispersion in toluene, after the in situ radical co-polymerization reaction. Other than that, a uniform PANi/PMMA blend is produced by using the co-precipitation method [40]. Since PANi has poor dispersibility, magnetite nanoparticles (for example, Fe_3_O_4_) have usually been added to improve the compatibility of PANi with PMMA. Magnetic nanoparticle is soluble in water, while PANi is hydrophilic in nature. Fe_3_O_4_ is easy to agglomerate and makes it difficult to adsorb analysts due to its surface simplicity. However, the large benzene ring structure in PANi enables a strong connection to Fe_3_O_4_ structure and hence acts as a stabilizer for Fe_3_O_4_. With the assistance of cellulose-based surfactant, a homogenous solution of PMMA/PANi/ Fe_3_O_4_ solution can be obtained. Not only that, the magnetite nanoparticles were also used to improve the thermal stability of PMMA. It was proven by the thermal analysis that the addition of PANi/ Fe_3_O_4_ limits the polymer hybrid weight loss at a temperature of as high as 600 °C. Generally, the addition of iron nanoparticles and cellulose surfactant, hydroxypropyl cellulose (HPC) created uniform dispersion between the PMMA and PANi. Stearic mechanism stabilization of HPC produces a stable colloidal dispersion between PMMA, PANi, and Fe_3_O_4_. In addition, strong interfacial bonding between the Fe_3_O_4_ with PMMA leads to better thermal stability. Similarly, Tomar et al. (2019) blended PMMA with PANi to overcome the poor interaction and solubility issues of PANi with the organic solvents [41]. The polymer blends were prepared by incorporating different PANi loadings of 0.4, 2.0, and 10 percent by weight (wt%) into a PMMA containing chloroform dispersion. In this work, iron chloride (FeCl_3_) was used as an oxidant to partially oxidize the PANi into a conducting polyaniline salt state.

## 3. Application of Conducting PMMA

The use of insulating polymers such as PMMA with conducting polymers such as PANi has proven to produce a new composite material not only with good mechanical properties but also with good electrical conductivity [42,43,44]. Thus, processable conducting PMMA has led to the development of a vast number of applications (Table 1).

### 3.1. Electrical and Electrochemistry

The high PANi electrical properties and the high PMMA mechanical features, when combined, can be useful in many applications, such as for the development of optical [44], electronic and optoelectronic devices [41]. The potential of PMMA/PANi cathode material as a substitution to metal was studied by Jia et al. (2017). Analysis done using linear sweep voltammetry (LSV), chronoamperometry (CA), and cyclic voltammetry (CV) proved that PMMA/PANi exhibited good mechanical stability with low background current density to surpass difficulties of cathodic polarization in metal when immersed in a marine environment [17]. The mixed solution containing both PMMA and PANi solution was coated on a glass carbon electrode to investigate their cathodic polarization. The results obtained indicated that the antifouling effectiveness of PANI-PMMA was outstanding when polarized from −0.4 V and −0.6 V several times, therefore making it possible to be used as the cathode in cathodic polarization [17]. The structural, optical, thermal, and conductivity properties of pure PMMA/PANi blends were investigated by Abutalib et al. (2019) to study the suitability of PMMA/PANi composites for electrochemical device application. However, graphene oxide (GO) was added into the blends to enhance the structural, optical, and electrical properties of PMMA/PANi prepared. It was found that the addition of GO into the blends reduces the crystallinity and optical band gaps of the composites. However, the thermal stability and electrical conductivity of PMMA/PANi composite was improved with increasing amounts of GO due to the increase in mobility of the charge carrier. The improvement of these properties with the addition of GO into PMMA/PANi composite is useful for electrochemical device applications [45]. Jankowska et al. (2020) produced PMMA/PANi electrospun fibers from the electrospinning process to be used as a support for laccase immobilization in dye decolorization. The suitability of PMMA/PANi electrospun fibers in laccase immobilization is due to the presence of numerous chemical functional groups on the surface of PMMA/PANi, which promotes successful immobilization of biomolecules by both adsorption and covalent binding methods. Despite the increased number of functional groups, PMMA/PANi electrospun fibers exhibited high porosity, which caused more laccase to be immobilized in 110 mg/g and 185 mg/g for systems with the enzyme attached by adsorption and covalent binding, respectively [46].

PMMA/PANi blends may also be a promising material for electrical application such as polymer light-emitting diode (PLED). The potential of PMMA/PANi blends such as PLEDS comes from the PMMA (insulator polymer) that acts as a donor while PANi (conducting polymer) acts as an acceptor. Not only that, but blending of PANi with PMMA has been found to improve luminescence, which is essential for PLED applications. However, optimization of several factors such as polaron-exciton quenching, donor concentration in donor-acceptor Förster resonance energy transfer (D-A FRET) system, branching of polymer chains, and, bipolaron states must be taken into consideration to obtain optimized luminescence of PMMA/PANi blends [18].

### 3.2. Coatings

Corrosion-protective coatings are normally applied on the surface of metals to prevent corrosion due to their cost-effectiveness. PANi has been used for anti-corrosion coatings mainly due to its good conductivity, better environmental stability, and good redox reversibility [48]. However, the limitations of PANi include poor adhesion and mechanical stability, which leads to the formation of pores that serve as a path for corrosion ion to penetrate the coating and cause corrosion. Thus, the performance of the coating can be improved by combining PANi with other polymers such, PMMA [40,48]. The works confirmed that blending of PMMA with PANi gave the best corrosion property with 99.95% corrosion protection even after immersion in 3% NaCl solution for 720 h. This excellent corrosion performance is due to the PMMA, which increases the degree of crosslinking of coating and thus reduces the porosity of the coating [48]. Similar research was conducted by Jia et al. (2018), in which electrospun PMMA/PANi microfibers coating was produced by deposition PMMA/PANi microfibers on carbon steel by electro-spinning and drop-cast methods. The results indicated that the hydrophobicity of PMMA enhances adhesion ability and reduces porosity, which leads to better barrier properties of the coating [49].

### 3.3. Sensing

PANi exhibits controllable electronic properties and room temperature sensitivity due to its variable oxidation states. Thus, PANi is an especially suitable material in sensing applications [50]. Saad Ali et al. (2020) fabricated a reliable and stable sensor made from PANi and PMMA for ammonia detection in a humid environment. This work reported that the PMMA material in the sensor solves the solubility difficulties and mechanical instability of PANi, therefore yielding a sensor with repeatable responses and good long-term stability in a humid environment [36].

For gas sensing applications, PANi electrospun fibers have been commonly used for the detection of various toxic chemicals including carbon dioxide (CO), hydrazine (N_2_H_4_), and ammonia (NH_3_). However, the commonly prepared PMMA/PANi electrospun fibers were in the form of nonwoven mat with different porosities, which resulted in inconsistency of the results obtained. As an alternative, a single fiber sensor with aligned PMMA fibers was demonstrated by Zhang et al. (2014). The prepared aligned PMMA/PANi fibers were reported to be higher in sensitivity and faster sensing response than the PMMA/PANi non-woven fibers sensors due to the larger surface area [21].

### 3.4. Medical

Actuators or artificial muscles are devices that mimic natural muscles. This is done due to external stimuli from voltage, current, pressure, or temperature that cause artificial muscles to rotate, expand, contract, and change their stiffness. Conducting polymers such as PANi are suitable to be used for actuators due to the reversible chemical reaction of PANi that enables permeation or release of cations from an electrolyte by applying an electric potential. Due to this, the conformational changes of the polymer chains are possible. PANis also have the ability to change their color simultaneously with the dimensional modification. As for smart artificial muscles, Beregoi et al. (2016) reported electrochromic properties of PANi-coated PMMA fiber webs that exhibited reversible and repeatable color change when the applied potential was continuously switched from 0 to 1 V and vice versa [23,24]. It was also found that the prepared PANi coated PMMA fibers possess a high degree of adhesion and good compatibility with eukaryotic cells when investigated by in vitro biocompatibility tests using cultured human amniotic fluid stem cells [23]. These results proved that metalized polymer fiber webs with good transparency such as PMMA is good to be used as electrode for PANI shell deposition by electrochemical polymerization. The color changes of PANi during oxidation state can be very easily changed by simply controlling the potential applied on the resulting structures in the presence of an electrolyte [23]. In summary, the suitability of PANi for actuators derived from the ability of PANi to change into various shapes and morphologies, which enhance the properties of the actuators [24].

## 4. PANi-Based Blends and Composite

Polymer blending refers to the process of mixing one or two different polymer types to create a new material. The process involves the physical mixing of at least two homopolymers to produce a new material with different properties in a short production time and with cheaper production cost [51]. To date, the literature has demonstrated that the polymer blends based on PMMA and PANi create a material with the synergistic combination of high mechanical properties from PMMA and high electrical conductivity from PANi [30,35,52]. Many methods, including the solution cast method, interfacial polymerization, and electrochemical polymerization, have been employed to produce the PMMA/PANi blends.

### 4.1. Solution Casting Method

The solution casting method is relatively easy to conduct and requires less reaction time [53]. The approach involves polymerizing the PMMA with benzoyl peroxide initator (BPO) until the mixture becomes viscous. Then, toluene is added into the PMMA before mixing with PANi and 1.0 M HCl [30]. In this way, homogenized blends can be obtained, since the viscous PMMA is diluted when dissolved in toluene. Another work on the synthesis of PMMA/PANi blends was employed by Kumar et al. (2020) by using hydrochloric acid (HCl) as dopant and oxidant potassium dichromate (K_2_Cr_2_O_7_) to facilitate the polymerization [53]. The blends were prepared using the solution casting method and with different weight percentages of PANi, namely 0, 0.5, 1.0, 5.0, and 7.0 wt%, which were added into a PMMA dichloromethane-containing solution. The solution was left stirring for 3 h before it was poured into a petri dish and dried for another 12 h to yield a composite membrane film. Another modification was reported by Fattoum et al. (2012), which used dichloroacetic acid as a solvent to prepare the PMMA/PANi film. The respective solutions containing different weight percentages of PANi powders (between 0.05% to 0.5%) were stirred for several hours at 60 °C before casting on a glass substrate. The PMMA/PANi film was produced by drying the solvent at 60 °C ambient temperature for several days to evaporate the solvent. The analysis performed using the UV-Vis spectrometer revealed two peaks at the 285 and 443 nm wavelengths, and these peaks were attributed to the π–π* transition and polarons transition of polyaniline chain [54]. A similar approach was done by Dimitriev et al. (2015), but this study focused on different concentrations of PANi and determined the effects on the morphology, electronic absorption spectrum, and the ability for acid doping. Moreover, this approach was used to confirm the nature of a cast film that depends critically on the relative rate of solvent evaporation with respect to the changes in the rate and subsequently the molecular interaction that occurred [55].

### 4.2. Interfacial Polymerization

Interfacial polymerization occurs at the interface between the two immiscible phases. This process is used to polymerize polymer particles and membranes and is one of the most effective methods to yield high-quality PANi in large quantities [56]. In this polymerization process, the monomers diffuse to the interface due to the differences between the chemical potentials of two immiscible liquids [57]. The benefits of PANi interfacial polymerization include better growth control of polymeric structures to produce a polymer with more complicated nanostructures [58].

A series of polymer materials including the particles, capsules, and membranes are prepared via the interfacial polymerization approach and are widely used in many electronic applications. Subsequently, the interfacial and rapid-mixing polymerizations were employed to produce the PMMA/PANi blends. In order to improve the poor solubility of PANi, Abu Talib et al. (2019) dissolved nanofiber polymer blends consisting of PMMA/PANi (80/20 wt%) in chloroform for 24 h at 50 °C for complete dissolution [45]. The characterization of the resulted polymers using FTIR analysis confirmed the interaction of PANi with the PMMA matrix, as the band that is attributed to the stretching vibration of N–Benzenoid–N and N=Quinoid=N at 1485 cm^−1^ and 1144 cm^−1^, respectively, was observed [45]. These wavenumbers are the signature bands of PANi, which refer to PANi benzenoid and quinoid moieties. The presence of these wavenumbers in the PMMA/PANi spectrum prove the formation of PANi in the PMMA matrix.

Zheng et al. (2018) carried out low-temperature interfacial polymerization of PANi core-shell microspheres. The polymerization temperature significantly affects the reaction rates of polymerization [59]. The polymerization of aniline monomers was initiated on the surfaces of microspheres via the interfacial polymerization reaction at 0 °C, with the APS initiator to start the polymerization through the seed-induced heterogeneous nuclear process [59]. A low temperature of polymerization resulted in the uniform coating of PANI shells on the surfaces of nanospheres, due to the low reaction rate and the absence of homogenous nuclear reaction of aniline [59].

Different types of oxidants, such as ammonium persulfate (APS), ferric chloride (FeCl_3_), and potassium dichromate (K_2_Cr_2_O_7_), were used in the interfacial polymerization of PANi and by using HCl as a medium [56]. This work reported that the formation of PANi in the presence of composite oxidants (APS/K_2_Cr_2_O_7_) becomes faster, as compared to the formation with a single oxidant. The diffusion of PANi monomers into the aqueous phase occurred due to the hydrophilic property of the monomers [56].

## 5. PANi Synthesis

### 5.1. Acid Dopants for PANi Synthesis

An acid dopant is usually used during the synthesis of PANi to improve its conductivity. One of the most common dopants used is hydrochloric acid (HCl) [60,61]. Sulphuric acid (H_2_SO_4_) is another acid dopant that is used as an alternative to HCl in the PANi synthesis. Banerjee et al. (2018) used the low molarity of strong H_2_SO_4_ (0.3 M) acid to improve the electrical conductivity of PANi, due to the increase in protonation ions [62]. The produced PMMA/PANi precipitation was then washed several times with acetone and deionized water before the drying process to get rid of the unreacted monomers and other byproducts. Doping is important in the PANi synthesis, as it is one approach to enhance the solubility, infusibility, processability, and poor strength of PANi [63]. These improvements are useful for developing PANi in many applications, including gas sensors, photovoltaic cells, supercapacitors [62,64], and lithium-ion batteries [65,66]. Other than HCl and H_2_SO_4_, malic acid (MA) was used by various researchers as a doping agent for PANi synthesis [67,68]. Hamlaoui et al. (2020) used maleic acid (MA) as doping and to create compatibilizer between the PMMA-PANi blends [67]. The acid diluted with distilled water was added by a dropwise addition method into the mixture containing PMMA/PANi solution and before the oxidant addition. The polymerization was conducted in two different phases like the bisphasic organic and organic phases. During interfacial oxidative polymerization of PANi, the rearrangement of internal distribution charges and spin between the PANi structure occurs. These changes create potential doping sites between the PANi protonated chains and acid. Nevertheless, Hu et al. (2016) synthesized PANi without the use of dopant during the synthesis process. This novel route created a cabbage-like PANi structure that was executed by the in situ polymerization of aniline monomers, as only the oxidants and hydroxylated PMMA nanospheres were available as a template. This new morphology of cabbage-like PANi can be potentially used as a supercapacitors due to the larger surface areas and π–π conjugated system scales [69]. However, in fact, the different types of dopant acids result in different redox states and doping degrees that can substantially affect the electrochemical properties of PANi [70]. Table 2 shows the different types of doping acids used by various researchers in the PANi polymerization.

The PANi morphology changes with the changing molarity of acids [77,78,79,80,81,82,83]. The self-assembled PANi nanotubes with an outer diameter between 100 to 150 nm and a 40 nm average wall were obtained when 0.1 M HCl was used [79]. In addition, the PANi nanotubes were observed to be rarely obtained from a mixture of strong acid/aniline solutions, while the increase in acid molarity led to the mixture formation of PANi nanorods and nanotubes [79]. Table 3 shows the difference of PANi micrographs obtained from different molarity of acid.

Zhang et al. (2018) further investigated the effects of acid concentrations on the PANi structure and reported that the PANi nanorods diameter produced from the oxidative chemical polymerization was estimated to be in the 130–150 nm range and with the length of several micrometers when HCl concentrations between 0.4 to 0.6 mol/L were used. In addition, the PANi nanorods synthesized using low HCl concentrations were found to contain some microstructures with rough surfaces. The diameter of PANi continued to decrease to 120 ± 10 nm when the HCl concentrations were increased to between 0.8 and 1.0 mol/L. The PANI structure was worm-like, with rough and uniform surfaces. However, the PANi structures with a smooth surface and a thinner diameter were obtained when a 1.2 mol/L HCl solution was used [84].

Numerous efforts have been made to synthesize PANi with various morphologies, scales [85,86], and shapes [87]. Apart from using different molarities of acids, the difference in PANi structures can be achieved by using various polymerization methods, such as electrochemical, [88] interfacial, [56] rapid mixing, [89], and emulsion polymerization methods [27].

### 5.2. Emulsion Polymerization Method

Among all the polymerization methods available for PANi synthesis, the emulsion polymerization method may be the most favorable due to the ease in nanoscale synthesis and in controlling the polymerization conditions [90,91,92,93]. The PANi particles produced through a reverse microemulsion method are spherical, with less than 100 nm in size [94]. The emulsion polymerization method is proven to be simple and fast to complete compared to other techniques [95,96]. During the emulsion polymerization of PANi, Chajanovsky et al. (2020) coated the nanoparticles with polyaniline (PANi) to produce a core-shell structure that was confirmed by high-resolution scanning electron microscopy (HRSEM). The analysis through HRSEM (Figure 3a,b) revealed that the PANi/nanoparticles membrane pore sizes are between 100–200 nm, with an average size of approximately 119 ± 28.3 nm [95].

Singu et al. (2018) employed a similar method and fabricated the nano-composites film consisting of PANi and multiwalled carbon nanotubes (MWNT). The characterization of the film using TEM analysis indicated that the MWNT diameters were about 100 to 120 nm when measured, while the sidewalls were observed to be smooth [97]. The FE-SEM image showing the as-synthesized pure PANi revealed the presence of nanofibers with diameters ranging between 20 and 30 nm. The significant changes can be seen in the morphology of PANI-MWNT-4 nano-composite, produced by the formation of PANi and MWNT. The highly uniformed PANI corals were present on the MWNT surface, which caused the roughness on the MWNT sidewalls. The difference in the structure of pure PANi, MWNT, and PANi-MWNT-4 nanocomposite is shown in Table 4. In addition, the diameter of the PANi-MWNT-4 nano-composite was found to be higher (between 150 to 300 nm) than that of MWNT (between 100 to 120 nm) and PANi nanofibers (between 20 to 30 nm). The length of thick PANi layers was determined to be several micrometers and hence proved the successful growth of PANi corals on the MWNT developed by the in situ emulsion polymerization method [97].

### 5.3. Polymerization Temperature

Polyaniline is normally synthesized through the oxidative polymerization method by using acid as the reaction media. However, this polymerization reaction is exothermic [98], and thus there is a need to control the polymerization temperature, as temperature has a prominent effect on the PANi structures and properties [68,94]. There have been several studies that reported the influences of polymerization temperature on the molecular weight and conductivity of PANi [99,100,101,102]. It was found that high molecular weight and crystallinity of PANi were due to the low polymerization temperature used during the synthesis process [103]. However, it was also reported by Yilmaz et al. (2009) that the polymerization temperature has no significant effect on the conductivity of PANi [103]. However, contradictory findings were reported by the other researchers that demonstrated that the conductivity of PANi increased with the decreased polymerization temperature [104,105]. Maity et al. (2016) reported the increment in PANi conductivity from 0.5 S/cm to 11 S/cm, with an increase in the polymerization time and a decrease in the polymerization temperature. The increase in conductivity may be attributed to an increase in the crystallinity and molecular weight [104]. At low temperatures, PANIi tends to grow slowly, thus improving the crystallinity formation that finally leads to the enhancement in electrical conductivity. Apart from this, it was also reported that the polymerization time has a significant effect on the thickness and sheet resistance of PANi films. Park et al. (2018) reported that the film thickness drops significantly from 625 nm to 269 nm and 200 nm when the polymerization temperature increased from −10 °C to 20 °C and 40 °C, respectively. At the polymerization temperature of 20 °C, the electrical conductivity was inversely proportional to both the thickness and resistance, with the conductivity value measured at 15.7 S/cm. However, these conductivity values were reported to be the highest, as compared with the PANi prepared using other polymerization temperatures [106].

### 5.4. Polymerization Yield

The polymerization yield is one of the crucial factors in determining the polymerization efficiency during the polymer synthesis process. Besides the monomer-initiator concentration and polymerization time factors, the types and molarity of acids used during the polymerization process may also affect the polymerization yield of PANi [92]. A study from Noby et al. (2019) indicated that increasing molarity of acids resulted in decreasing polymerization yield. This might be due to the slow polymerization rate of PANi in highly acidic media [79].

The effects of oxidant amount, aniline amount, and H_2_SO_4_ amount on polymerization yield are tabulated in Table 5 [92]. It is evident from Table 5 that the yield of PANi increases with the increase in the oxidant amount in the range of 0.25 to 1.25 mmol. However, the excess addition of oxidant caused the retarding effect on the polymer yield. The retardation process is caused by the over-oxidized radical cations, resulting in growth rate declines and shortened polymer chain length. Apart from this, a low concentration of radical cations will also lead to low polymer yield at a low oxidant amount of oxidant [92]. In terms of the monomer concentration, the highest yield of 10.95 mmol was recorded with the highest concentration of aniline. However, as expected, beyond this optimum amount of 10.95 mmol, the oxidant efficiency began to drop due to the excessive monomer amount to oxidize. Similar trends were also reported from the other researchers, as these studies agreed that too many monomers led to the reduction in polymerization yield [107,108,109,110]. The reduction was due to the incomplete conversion of aniline monomer to PANi [107] due to insufficient oxidation [110]. In the case of the relationship between the acid concentration with polymerization yield, the improvement in yield was noted when the amount of acid was increased from 3.75 to 12.5 mmol. A similar trend was observed when the excess of acid led to a reduction in polymerization yield [110]. High polymerization yield with a high acid amount is due to the high electrostatic repulsion within the PANi chains, caused by better-extended conformation [92]. Nevertheless, the hydrolysis of PANi chains occurs at a higher amount of acid, causing the yield of PANi salts to decrease [111].

## 6. Co-Polymerization of PANi with Various Thermoplastics

### 6.1. Graft PANi Copolymer

Co-polymerization refers to the polymerization of more than one monomer, yielding a copolymer with better controls and functionality. Grafting is one type of co-polymerization that produces a copolymer consisting of a branched molecular structure and with a chemical composition that is different from the backbone. It can be produced with one or more molecular chains, while the linear backbone may have multiple branches (the grafts) formed from the different macromolecular chains [112]. Since PANi is incompatible, it is not easily blended with most organic solvents [113]. The aniline co-polymerization by grafting onto the polyvinyl monomers or its copolymers can be performed to improve the PANi solubility in organic solvents [114]. Abbaspoor and his co-workers demonstrated that the copolymers of PANi can be grafted with amorphous polymers by using the interfacial polymerization approach [115]. The polymerization involved a self-seeding technique to grow the PANi/PMMA mixed-brush single crystal with a common crystalline block of polyethylene glycol (PEG). The characterization was carried out using a ^1^HNMR spectrometer to identify the chemical structure, while a GPC was used to calculate the molecular weight of the produced copolymer. Another graft co-polymerization of PANi with other monomers was carried out by Abd el-Mageed et al. and his co-workers [114]. Such polymerization involved the co-polymerization of aniline with the thermoplastic monomers in an acidic medium with APS as the initiator at 40 °C for 2 h under a nitrogen gas atmosphere. The analysis was done by using the UV-Vis spectrometer to confirm the grafting of PANi onto the polymer matrix. The results showed peaks that are attributed to PANi at 275 nm and 536 nm. The peaks were translated as the π–π* transition of the benzenoid segment and the bipolaron transition in polyaniline [114].

In addition, the chemical modification through graft co-polymerization has been applied to overcome poor processability of PANi [116,117,118]. Alizadeh et al. (2015) prepared the PANi grafted with thermoplastic, through the thermoplastic atomic transfer radical polymerization (ATRP) on the backbone of polyaniline macro-initiators. The thermoplastic polymers were chosen as a matrix to support PANi, as these polymers are easy to process, vulnerable to modification, and have diversely tuneable physical and chemical properties [119]. The polymerization was done under an inert atmosphere and by using dimethyl sulfoxide (DMSO) as a solvent. The copolymer was left to polymerize for 72 h at 25 °C, followed by the washing process using methanol. Finally, the products were freeze-dried to collect the copolymer precipitate.

### 6.2. Block PANi Copolymer

Block copolymer is a type of copolymer that consists of two or more covalently bonded block polymers. Among all the other techniques used for the thermoplastic modification of PANi, block copolymers are preferred due to their ability to control and tune the uniform PANi brushes distribution [120]. Nazari et al. (2018) fabricated conductive rod nanobrushes of PANi from the thermoplastic block PANi copolymers. The thermoplastic block PANi copolymers were synthesized by the interfacial polymerization process and through the use of ammonium persulphate, APS, and potassium hydrogen diiodate (PHD) as an oxidant [121]. The results indicated that the PANi synthesized using the PHD antioxidant is longer in chain lengths and has narrower diameter distribution and higher conductivity. Table 6 summarizes the advantages and disadvantages of grafting and block copolymer in terms of PANi copolymers.

Co-polymerization enables a good combination of properties between two different monomers or the production a new polymer with attractive new properties [131]. However, the main challenge of co-polymerization is to avoid the random distributions between the copolymers along the polymer backbone [127]. To overcome this, Chotsuwan et al. (2017) designed a copolymer of PANi and polythiophenes by employing the triblock (A-B-A) system. It was found that the stability of the triblock copolymers under visible light irradiation at ambient temperature was improved by the addition of PANi block [127]. It was also reported by the previous literature that the triblock copolymer has better control of the on-chain orientation and thus is able to give an improvement in the produced conjugated polymers properties [60]. Table 7 summarizes the advantages and disadvantages of the copolymerization and blending methods.

## 7. Conductivity

### 7.1. Doping

It is undeniable that the conductivity of PANi mainly depends on the dopant ions applied. Doping may increase the conductivity of semiconductors to become more conductive [139]. There are two types of doping, n-doping and p-doping. The conduction in n-doping is due to the movement of additional electrons, while the conduction in p-doping is due to the movement of positively charged holes [140]. Bandeira et al. (2020) compared the electrical responses between the undoped and doped PANi with different weight percentage, wt% of benzoic acid (between 0 to 25 wt%). In their work, they reported that the conductivity increases linearly with the doping level, with the maximum conductivity of 1.68 × 10^−4^ (Ω·m)^−1^ was measured at the maximum loading on benzoic acid attempted, at 25 wt%. The conductivity of the doped PANi is proposed to be governed by the p-type character through converting the PANi in emeraldine-based formed (PANi-EB) into the emeraldine salts form (PANi-ES). Moreover, the protonation by benzoic acid caused the counter ions interaction within the intra- and inter-chains of PANi, affecting the delocalization and modifying the electron mobilities, thus improving the conductivity [141]. Zhang et al. (2018) fabricated PANi with worm-like structures and found that the more stable structures of PANi with the highest conductivity were produced when synthesized using the higher HCl molarity [84]. They also stated that the increase in conductivity might be due to the high density of electrons and the decrease in the PANi diameter [84]. These findings were also supported by the other researchers, as the morphology of PANi affects the electrical conductivity of π-conjugation polymers due to the arrangement of PANI molecular chains and density of electrons [142,143]. This can be further explained by the movement of the electrons that travel throughout the worm-like nanorod length of PANi and further expand to the branched knobbles. The movement results in enhanced delocalization of charges, so it is advantageous for electrical conductivity enhancement [84].

Even though acid doping is responsible for improving the conductivity of PANi, the further increase in the HCl concentrations also leads to a decrease in the conductivity [84]. A similar result was obtained by Rahayu et al. (2019), who observed that the PANi soaked with different levels of HCl amplified the conductivity before the addition of subsequent concentrations decreased the conductivity value (Table 8). This was caused by the damages to the polymer chain structures in high acid concentration [144].

Even though acids were known to be good dopant to protonate PANi, other types of dopants such as graphene [145], iodine [146], carbon nanotubes (CNTs) [147,148], iron [149,150], and transition metal [151,152] were also proven to be useful in improving the electrical properties of PANi. Even though PANi has been extensively used in the fabrication of supercapacitor materials, the PANi-based supercapacitors suffered from short life cycles. This resulted in the development of PANi composites built with carbon-based materials and transition metal oxides. In the research conducted by Mondal et al. (2015), graphene was used as a dopant and soft template for the graphene-PANi composite synthesis and propelled the PANi application as high-performance supercapacitor electrode materials. The results indicated that the graphene increased surface areas in nanotube morphology to give better conductive paths, essential for fast electron transport, and leads to maximum specific capacitance [145]. Meanwhile, the results obtained by Wang et al. (2015) indicated that the iodine dopant used in PANi synthesis efficiently enhances the PANi composite’s electrical conductivity and electrochemical performances, with 712.5 F g^−1^ at the current density of 1 A g^−1^ [146]. Research done by Elnaggar et al. (2017) attempted to compare the conductivity of graphene and CNTs doped with PANi. The study observed higher conductivity in graphene-doped PANi composites caused by higher surface areas of the graphene sheet [153]. Similarly, the results reported by Alghunaim et al. (2019) on the iron effects on electrochemical properties of PANi/graphene oxide composites indicated that the Fe^3+^ dopant serves as a charge transfer tunnel and increases the synergistic effects between the polyaniline and GO [150]. There are various types of transition metal ions, namely Cu^2+^, Zn^2+^, Fe^2+^, and Fe^2+^, respectively, that can be used as a dopant for PANi. Subsequently, PANi doped with different transition metal ions has recently attracted a great deal of attention due to its possible functional applications as a redox-active catalyst [154] and corrosion inhibitor [155,156]. It was found that PANi doped with the combination of iron and zinc dopants exhibited higher current density and enclosed area in cyclic voltammetry (CV) curve analysis, potentially due to the simultaneous presence of Fe^3+^ and Zn^2+^ ions in the nano-composite [152]. However, it was reported that the doping of PANi might not be sufficient enough to make PANi into useful and efficient electromagnetic interference (EMI) shielding materials. Thus, Zhang et al. (2019) co-doped PANi with HCl and DBSA by layering with transition metal carbonitride (Ti_3_C_2_T_x_) via ionic intercalation, followed by a sonication-assisted method. It was reported that the conductivity of the PANi composite films was increased with an increased Ti_3_C_2_T_x_ ratio, while the highest conductivity was measured at 24.4 S/cm for PANi film with a mass ratio of 7:1 (Ti_3_C_2_T_x_:c-PANI) [157]. Table 9 below shows the different conductivity measured when different dopants were used. Other researchers used other types of dopants, such as Sn(IV)iodophosphate (SnIP), to improve the conductivity of PANi [158].

According to the results presented in Table 9, it can be concluded that the highest conductivity was achieved when PANi was doped with lithium chloride (LiCl). From Figure 4, it can be seen that the electrical conductivity was increased from 14.96 to 25.01 S·cm^−1^ when the LiCl concentration was increased from 0.125 to 2.5 M. This improvement can be explained by the mobility of Li^+^ ions towards the doping sites present on the polymer chain, causing the nucleophilic doping reaction. With the increase in LiCl concentration, the counterions were doped effectively on the polymers’ surfaces, therefore improving the electrical conductivity [161]. However, in agreement with the previous literature, too much dopant concentration caused the conductivity to decrease, due to the saturation or overoxidation of the polyaniline backbone [14].

### 7.2. PANi/PMMA Ratio

PANi has good electrical conductivity (EC), a promising property that is important for industrial applications. The EC value of pure PANi without doping is as high as 12.99 S·cm^−1^ [161]. One of the alternatives to improve the physical properties of PANi, such as the mechanical strength and thermal stability, is to create composites/blends through the mixing with PMMA [45]. The DC electrical measurement using a four-point probe setup was employed by Francis et al. (2020) and revealed that the increase in PMMA ratio led to the reduction of conductivity, as the conductivity measured for PANi-PMMA blends synthesized in the 1:0.1 and 1:0.2 ratio were 1.72 and 1.38 S/cm, respectively [30]. However, since PMMA is known to exhibit high-strength thermoplastic, the compression strength of the PANi-PMMA blends was found to increase to 0.1168 N/m^2^, compared to that of pure PANi, with a reported strength of 0.0811 N/m^2^ [30]. Similar research was done by Martynyuk et al. (2020) using different PANi volume percentages (v%) from 0 to 100 v%. It was established that the specific conductivity in the PMMA-PANi polymer composites was increased by more than 9 orders of magnitude, as compared to the pure PMMA matrix with the lowest amount of conducting PANi [162]. Additionally, the conductivity enhancement of the PANi composite may be due to the continuous electrically conductive phase that is homogeneously distributed over the entire composite polymer volume. Accordingly, the conducting network within the PMMA host polymers may be responsible for the formation of this conductive phase [162]. The incorporation of different PANi concentrations (i.e., 1, 3, 5, 10, 15, and 20 wt%) into thermoplastic was carried out by Rozik et al. (2016). The findings based on the dielectric investigations reflected an increase in both the permittivity and dielectric loss with the increase in the PANi contents due to the interfacial polarization and conductivity effects. In addition, the rise in composite conductivity was attributed to the ease of the composites’ lower percolation threshold formation due to the improvement in conductive pathways between the fillers, hence resulting in the improved dispersion of PANi [35]. On the other hand, in terms of thermal analysis, the thermal stability of the PS/PMMA (50/50 wt%) blends with different PANi concentrations showed an increase in thermal stability, due to the presence of PANi [35]. Moreover, other researchers reported that the alternating currents, for example, AC conductivity (σ_ac_), dielectric permittivity (E_0_), and dielectric loss (tan d) of the PMMA/PANi nanocomposites were remarkably amplified with the PANi hybrid nanofillers addition into the PMMA matrix (Table 10). The improvement in the AC conductivity is due to the interconnected spatial networks of nanotube–PANi–nanotube within the PMMA matrix [163]. In conclusion, this modification broadens the synthesized polymers’ potential to be used in many dielectric applications [163].

It is undeniable that by increasing the amount of PANi, a higher improvement in PMMA/PANi composites’ conductivity is achieved [164,165,166,167]. However, many researchers agreed that the excess amount of PANi loading leads to the reduction of conductivity in the composite prepared [168,169,170]. Research done by Moussa et al. (2017) reported that the direct current (DC) conductivity increased by about two orders of magnitude by increasing the PANi loading from 10% to the saturation loading of 70%. The increment can be attributed to the improvement in charge density and electrical mobility when PANi is added to the polymer matrix [27]. As expected, the conductivity starts to decrease after the optimum PANi loading concentration is reached. The conductivity reduction is due to the restriction in charge carriers mobility of PANi, thus proving that the electrons mobility issue is a major contribution that affects the improvement in conductivity [27].

## 8. Morphology

The addition of PMMA into PANi modifies the surface roughness of PANi [171,172], thus proving the occurrences of blending and formation between the PMMA and PANi [45,173,174,175]. The morphological analysis reported by Abutalib et al. (2019) used a scanning electron microscope (SEM) to observe the PMMA/PANi blends. As tabulated in Table 11, the results from the SEM scanning revealed the smooth, transparent, soft, and uniform surface of pure PMMA, while the PANi micrograph was observed to be rough with few cracks. However, when the PANi was added into the PMMA, the cracks were less visible on the blends’ surface, proving the miscibility between the two polymers. The miscibility may be due to the formation of hydrogen bonding between the PANi and PMMA polymers [45].

The morphological analysis of PANi was also studied by Sahu et al. (2018) using transmission electron microscopy (TEM). This analysis proved that the strong aggregation of PANi is caused by the intermolecular hydrogen bondings of PANi chains. The microstructure of the PANi matrix changed with the addition of PMMA. Hence, the agglomeration tendency of the PANi matrix was reduced. Thus, it can be stated that the PMMA matrix acts as a continuous phase, while the PANi acts as a dispersed phase; thus, the self-aggregation of PANi matrix is fully prohibited [163]. The optical micrographs of pure PMMA and PANi/PMMA composites membranes with different wt% (1,5,7 wt%) of PANi were further studied by Kumar et al. (2020). It was observed that the agglomeration of PANI particles increased with the increase in PANi loadings that led to the formation of the grain. The agglomeration was even more noticeable when the PANI distribution in the PMMA matrix was non-uniform [53].

### PMMA/PANi Nanofibers

The PANi nanofibers are becoming the potential interest of sensor development field due to the higher degree of stability, the ease of synthesis, and the flexibility in controlling conductivity by altering protonation and oxidation processes, as well as being agglomeration-free [176]. Various techniques can be employed to produce the PANi nanofibers, namely electrospinning, phase separation, extrusion, and template synthesis approaches. By far the most favorable method is the electrospinning method, due to its simplicity and versatility; most importantly, the diameter of the nanofibers can be controlled [176].

Since the PANi polymer was not suitable to be electrospun into long-fiber, long-chain-length polymers, such as PMMA, were usually used to blend with PANi during the electrospinning process [20,22,177]. Figure 5 shows the SEM micrograph of the PMMA/PANi nanofibers. Another work of PMMA/PANi blend nanofibers were presented by Anwane et al. (2018) through the use of electrospinning and dip-coating polymerization techniques, where the PMMA nanofibers were inserted into the PANi solution containing aniline monomer and HCl. The resultant PMMA/PANi nanofibers can be potentially used as a quick sensor, as compared to the PANi thin film due to the higher surface area to volume ratio and larger aspect ratio [22].

Another work on the preparation of PANi nanofibers was carried out by Abdali et al. (2017). The study used reduced graphene oxide (rGO) nanofiller to further improve the electrical and thermal properties of PMMA/PANi blends [177]. In this work, rGO solution was added into the PANi and PMMA solutions to form a homogeneously mixed solution for the electrospinning process. It was found that there was an improvement in the thermal properties of PMMA/PANi nanofibers in the presence of graphene as the thermal degradation was increased to ≈441 °C, a magnitude higher than the nanofibers without graphene at ≈348 °C [177].

PANi/PMMA nanofiber can be potentially used as sensing material [178]. Vu et al. (2019) fabricated electrospun PMMA/PANi nanofibers by using PMMA that serves as a matrix for PANi in the electrospinning process due to the transparency and good stability of PMMA [20]. In addition, the prepared nanofibers underwent surface treatment using the UV/Ozone treatment to improve water molecules diffusion. The results obtained in this work are feasible in the fabrication of humidity sensors. Apart from that, the PANi/PMMA nanofibers with modified electronic structure can be practically used as a highly sensitive sensor for detection of volatile organic compounds (VOCs) [20].

## 9. Conclusions

Table 12 presents the research direction and its challenges and applications for PMMA/PANi as conducting materials. It can be said that many research studies have been carried out in fabricating conducting poly (methyl) methacrylate (PMMA) by incorporating polyaniline (PANi) either by physical blending or copolymerization. The main challenge is to produce new material that exhibits high conductivity but with a good mechanical property. There are great number of factors during PANi synthesis that may affect the conductivity of PANi. Specifically, different types and molarity of dopants resulting in different conductivity of PMMA/PANi have been produced. Overall, different dopant molarity and polymerization methods have also led to different structures of PANi. Among all of the polymerization methods, the emulsion polymerization method is considered to be the simplest method to polymerize PANi at nanoscale size. It should not be neglected that the ratio of PMMA and PANi may also vary the conductivity of the composite or copolymer produced, due to the increase of conducting network within the host polymers. PMMA is known as a high-strength material, including high tensile and compressive strength, making it the best candidate host material to enhance PANi performance. The high transparency and mechanical stability of PMMA can lead to the development of new conducting material that can potentially be used in many applications such as optoelectronics, sensors, actuators, corrosion protection, and many more.

Despite these advances in PMMA/PANi-based conducting materials, much effort must be made to produce homogeneous blends of PANi with PMMA to be useful for industrial use. Powdered PANi is known to be less soluble in most organic solvents, especially in its doped form. The addition of PMMA is proven to improve the solubility of PMMA/PANi composites, but leads to poor processing of the prepared composite. Thus, the addition of surfactants such as sodium dodecyl sulfate (SDS) is required to overcome this limitation.

High electrical conductivity is another criterion that needs to be taken into consideration. In fact, the conductivity of PANi is not very high. PANi in its undoped state was reported to have conductivity values at around 5 × 10^–4^ S/cm. Thus, modification is needed such as doping PANi with acid dopants or adding conducting filler such as graphene oxide (GO) and carbon nanotubes (CNTs). Dedoping of PANi with ferric chloride, FeCl_3_ is also proven to amplify PANi conductivity, in a which secondary doping leads to the structural rearrangement of PANi and hence increases the π conjugation as well.

In recent years, copolymerizing PMMA with PANi has led to the breakthrough of this copolymer material in various applications, especially as a functional material to be used in the electronics circuit, sensors, and solar cells. However, synthesizing highly crystalline polymer is still becoming the critical issue as copolymerizing PMMA with PANi reduces crystallinity of the prepared copolymer. In solar cells application, the prepared PANi copolymer needs to exhibit greater solubility, durability, and sustainability. On the other hand, poor solubility of PANi and low crystallinity of the resultant copolymer limits its suitability for this application. Thus, much work remains to be done for the optimization of the above physical property.

In spite of these challenges, we believe that the next generation of PMMA/PANi composites either by copolymerization or blending should meet the high conductivity and better mechanical stability standards as research progresses. Future research can be carried out in the following directions:

(1)For PMMA/PANi copolymer, the selection of the second monomer to copolymerize with PANi must consist in specific functional groups to incorporate into the backbone of PANi without reducing its crystallinity.(2)Solving the problem of incompatibility of the mixed polymer by optimizing certain parameters such as molecular weight and viscosity for both polymers and choosing the right solvent during mixing.

## Figures and Tables

**Figure 1 polymers-13-01939-f001:**
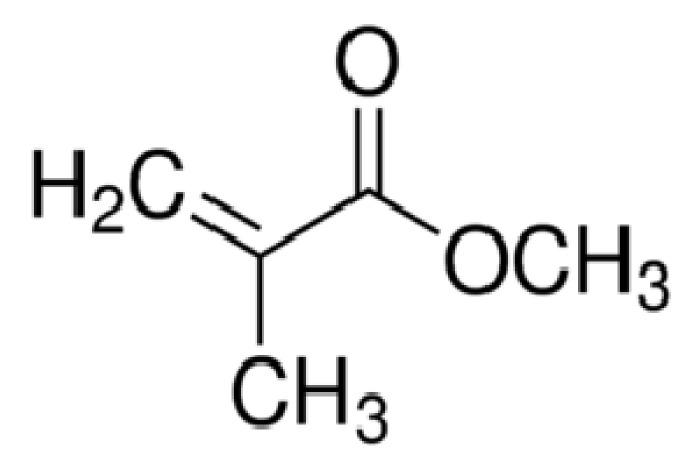
Methyl methacrylate (MMA) monomer.

**Figure 2 polymers-13-01939-f002:**
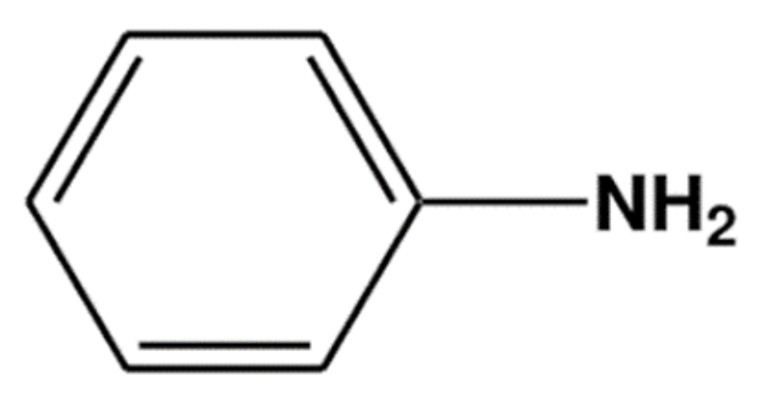
Aniline monomer.

**Figure 3 polymers-13-01939-f003:**
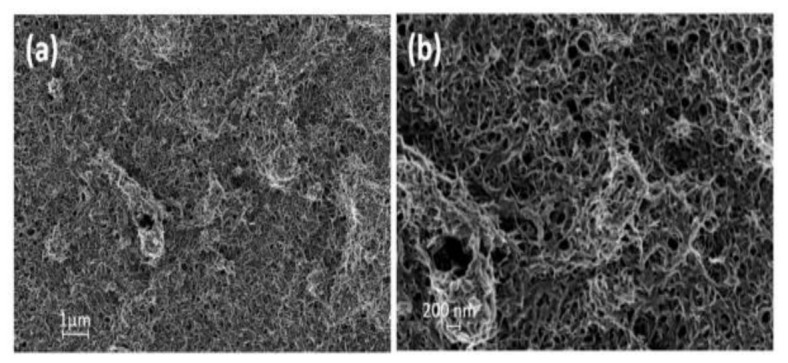
HRSEM images of the PANi grafted CNT sample showing (**a**) 1 µm membrane pore size, (**b**) 100 nm membrane pore size [95].

**Figure 4 polymers-13-01939-f004:**
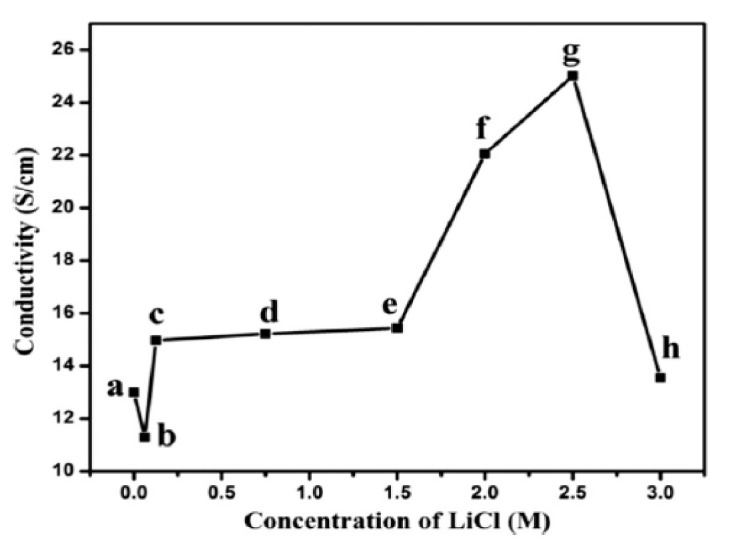
Electrical conductivity of PANI vs. concentration of LiCl (**a**) PANi, (**b**) PANi-LiCl (0.0625 M), (**c**) PANi-LiCl (0.125 M), (**d**) PANi-LiCl (0.75 M), (**e**) PANi-LiCl (1.5 M), (**f**) PANi-LiCl (2.0 M), (**g**) PANi-LiCl (2.5 M), and (**h**) PANi-LiCl (3.0 M) [161].

**Figure 5 polymers-13-01939-f005:**
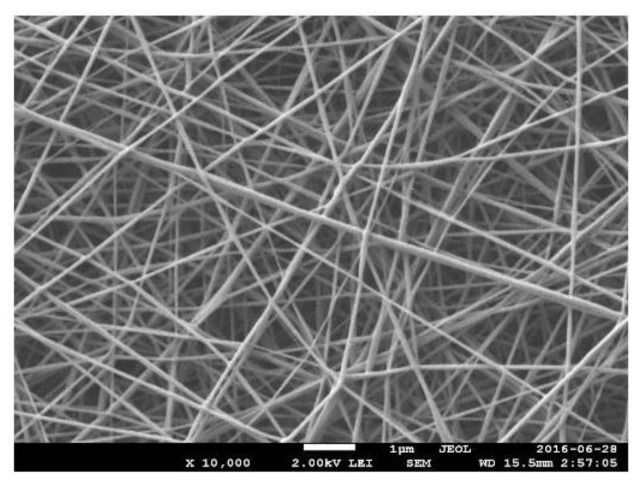
SEM micrograph of the PMMA/PANi nanofibers [177].

**Table 1 polymers-13-01939-t001:** Potential applications of PMMA/PANi blends and composites.

Applications	Material	References
Electrode Material	PMMA/PANi Coating	[17]
Polymer Light Emitting Dioeds (PLED)	PMMA/PANi Blends	[18]
Electrochemical Device	PMMA/PANi nanocomposites	[45]
Dye Decoloration	PMMA/PANi fibers	[46]
Packaging	PMMA/PANi/PVDF Blends	[47]
Humidity Sensors	PMMA/PANi fibers	[20]
Gas Sensor	PMMA/PANi Fibers	[21,22]
Smart Actuators	PANi-coated PMMA fiber webs	[23,24]
Corrosion Protector	PMMA/PANi Coating	[48,49]

**Table 2 polymers-13-01939-t002:** Acid dopants with various molarity use for polymerization of PMMA/PANi.

Type of Acid	Molarity (M)	References
Hydrochloric acid (HCl)	1.5 0.2	[71] [72]
Sulphuric acid (H2SO4)	1	[73]
Tartaric acid (TA)	1	[74]
Citric acid (CA)	0.001	[68]
Camphorsulfonic acid (CSA)	0.001	[75]
Malic acid (MA)	0.001	[76]

**Table 3 polymers-13-01939-t003:** Comparison of PANi structure produced from different molarities of acid [79].

Acid	PANi Structure	Thickness (nm)
0.1 M HCl	Nanotubes and Nanorods	≈40
2 M HCl	Nanoflowers and Nanofibers	≈40
5 M HCl	Nanosheets	≈40
1.1 M H2SO4	Nanofibers	-

**Table 4 polymers-13-01939-t004:** The structure and diameter of pure PANi, MWNT, and PANi-MWNT-4 [97].

Sample	Structure	Diameter (nm)
Pure PANi	Nanofibers	20–30
MWNT	Nanotubes	100–120
PANI-MWNT-4	PANi corals	150–300

**Table 5 polymers-13-01939-t005:** The influence of synthesis parameters on the percent yield of PANi–DBSA–H_2_SO_4_ salts [92].

Parameter	Polymerization Yield (%)
Oxidant (mmol)	
0.25	≈38.0
0.5	≈39.0
0.75	≈41.0
1	≈44.0
1.25	≈50.0
1.5	≈45.0
1.75	≈42.0
Aniline (mmol)	
1.09	≈50.0
5.47	≈57.5
10.95	≈70.0
16.42	≈62.5
21.9	≈50.0
27.38	≈37.5
H2SO4 (mmol)	
3.75	≈31.0
5	≈32.0
7.5	≈36.0
12.5	≈50.0
17.5	≈40.0
22.5	≈36.0
25	≈35.0

**Table 6 polymers-13-01939-t006:** Advantages and disadvantages of graft and block copolymers.

Type of Copolymer	Advantages	Disadvantages
Block	Most suitable for conducting nanostructured polymers ^1^	Difficult to control the length of nanostructure precisely ^3^
	Exhibit reversible or irreversible changes in physical properties or chemical structure (stimuli responsive amphilic block copolymers) ^2^	Produces highly oriented perpendicular to the substrate PANi nanostructure but high in porosity which might be useful as electrode material, but not as corrosion protection ^5^
	Ease of self-assembly morphology ^3^	Require slow annealing to improve the long range order of the block copolymer microdomains ^5^
	Contain microphase segregation that will promote intermolecular conductivity when doped ^4^	The monomer blocks are independent in their electrochemical response ^6^
	Easy to obtain ordered, uniform, and highly homogenous polyaniline nanostructure ^5^	The rate of bipolaron formation is slow when the number of the blocks increased ^6^
	Better control of polymer chain orientation ^6^	Bipoloron tends to trap in the PANi blocks ^6^
	Exhibit better optical and electrochemical properties than random copolymers ^6^	Require high doping levels ^3,6^
Graft	Simple and easy equipment to carry out (in situ polymerization) ^7^	Difficult to graft long chain of polymer onto the polymer backbone ^7^
	Better organization of chains even at lower PANi content ^7^	Free polymers are easily removed during washing ^7^
	Grafting copolymer transports holes more efficiently and lowers the leakage current ^8^	Higher PANi contents lead to the formation of aggregation ^7^
	Grafting can be prepared by both in situ polymerization and simple blending method ^9^	Rigid and long PANI-grafted chains decrease the ability of cross-linking reaction of the second monomer ^7^

Source: ^1^ Huang et al. (2010) [122]. ^2^ Hu et al. (2010) [123]. ^3^ Wang et al. (2021) [124]. ^4^ Shimano et al. (2001) [125]. ^5^ Kuila et al. (2010) [126]. ^6^ Chotsuwan et al. (2017) [127]. ^7^ Marcasuzaa et al. (2010) [128] ^8^ Jung et al. (2010) [129]. ^9^ Bae et al. (2004) [130].

**Table 7 polymers-13-01939-t007:** Advantages and disadvantages of copolymerization and blending methods.

	Copolymerization	Blends
Advantages	Can be carried out either by electrochemical or chemical polymerization ^1^	More convenient and lower cost than copolymerization ^3^
	Higher conductivity due to the lower percolation threshold ^1^	Require lower amount of PANi to achieve conductivity of the blends ^4^
	More homogeneous phase structure ^2^	Can be prepared by either in situ polymerization or simple solution blending ^2^
	Produce one phase structure ^2^	Solution blending contains higher content of PANi ^2^
Disadvantages	Morphology PANi changes from spherical particles to uniform and non-particulate structure as PANi content increases ^2^	Difficult to produce homogenous blends due to insolubility of PANi in common solvents ^1^
	Atom transfer radical polymerization (ATRP) technique is required for controlling radical polymerization ^5^	Require high acidic PANi to solubilize PANi ^1^
	Choices of monomers are limited ^6^	Produce two-phase structure ^2^
	Radical polymerization is light- and moisture-sensitive ^7^	Lower melting point (T_m_) and glass transition temperature (T_g_) ^3^
	Most ionic polymerization need to be carried out under inert environment ^8^	Immiscible-Limited its potential for certain applications ^4^

Source: ^1^ Bae et al. (2003) [132]. ^2^ Marcasuzaa et al. (2010) [128]. ^3^ Albertsson et al. (1997) [133]. ^4^ Stockton et al. (1993) [134]. ^5^ Gheybi et al. (2007) [135]. ^6^ Edmondson et al. (2017) [136]. ^7^ Chen et al. (2016) [137]. ^8^ Su et al. (2013) [138].

**Table 8 polymers-13-01939-t008:** Conductivity measurement of PANi synthesized with various HCl molarity [144].

HCl Molarity (M)	Conductivity (S·cm^−1^)
0.25	4.34 × 10^−2^
0.50	7.33 × 10^−2^
0.75	3.31 × 10^−2^
1.00	7.08 × 10^−2^
1.25	6.27 × 10^−2^
1.50	1.66 × 10^−2^

**Table 9 polymers-13-01939-t009:** Conductivity measurement of PANi synthesized with various types of acid dopants.

Dopants	Conductivity (S·cm^−1^)	References
HCl	9.3 × 10^−1^	[159]
H_3_PO_4_	5.6 × 10^−1^	[159]
Graphene	1.2 × 10^−1^	[153]
CNTs	1.2 × 10^−2^	[153]
Iodine	4.8 × 10^−1^	[146]
Copper	2.9 × 10^−1^	[160]
LiCl	25	[161]

**Table 10 polymers-13-01939-t010:** AC conductivity (σ_ac_), Dielectric Permittivity (E_0_), and Dielectric Loss (tan d) of PMMA with various amount of PANi hybrid nano-composites [163].

PMMA/PANi Ratio	AC Conductivity,σ_ac_ (S/cm)	Dielectric Permittivity, E_0_ (έ)	Dielectric Loss (tan d)
Pure PANi PMMA/0.101 wt% PANi	≈10^−2^ ≈10^−6^	10^3^–10^4^ 10^1^–10^2^	10^0^–10^1^ ≈10^−2^
PMMA/0.101 wt% PANi@AgNP	≈10^−5^	≈10^2^	10^−2^–10^−1^
PMMA/0.101 wt% PANi@f-CNT	≈10^−1^	≈10^4^	≈10^1^

**Table 11 polymers-13-01939-t011:** Comparison of structure and morphology properties between pure PMMA, pure PANi, and PMMA/PANi blend.

Sample	Structure and Morphology Properties	
	References	
	[45]	[163]
Pure PMMA	Transparent, soft, uniform (smooth surface)	-
Pure PANi	Nanofiber morphology (rough surface)	Globular microstructure, strong aggregation
PMMA/PANi Blend	No cracks (Improved surface roughness)	Low aggregation

**Table 12 polymers-13-01939-t012:** Summary of conductivity studies for conducting PMMA/PANi.

Summary	Remarks
Factors affecting Conductivity	Types of dopants Molarity of dopants PMMA and PANi ratio Addition of filler Method of synthesizing: Copolymerization or chemical blending
Challenges	Producing new material that exhibits high conductivity but with good mechanical properties Producing homogeneous blends of PANi with PMMA Maintaining crystallinity of the copolymers as copolymerization reduces crystallinity
Future Directions	Modification of PANi structure by copolymerization with a suitable monomer to overcome limitations of PANi Optimization of PMMA/PANi properties to fully exploit the outstanding properties of both PMMA and PANi
Applications	Electronics and Optoelectronics Sensors Actuators Corrosion Protection

## Data Availability

The data presented in this study are available on request from the corresponding author.

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
