# Peer review of "Synthesis and Conductivity Studies of Poly(Methyl Methacrylate) (PMMA) by Co-Polymerization and Blending with Polyaniline (PANi)"

_polymers, 2021, doi:10.3390/polym13121939_

Round 1

Reviewer 1 Report

The review article is well written and presents the current state of the art on this highly interesting topic. The review focuses on different approaches for incorporation of PANi as a conducting material into PMMA. Different dopants and changes in surface structure and morphology are discussed and reviewed. The only drawback in my opinion is that there is little emphasis given on its application- for example the importance of these materials in sensing or even medical application should be added. It would be good to correlate some of the sections- like morphology with its application- nanostructured morphology may significantly influence on sensing as well as biological response. For example, PMMA material is also known for its good biocompatibility thus discussion on its application in medical filed would be valuable. Overall the manuscript gives full review on different approaches and I would suggest the article to be published after minor revision.

  • On page 2, line 78- please add “..PANi has been incorporated with PMMA for various applications…which applications with references
  • ) Table 2- correct HCL to HCl in the table title
  • Correct typo probably PAni to PANi- page 11, line 404 and 405? And line 413 PANI- to PANi?

Reviewer 2 Report

The aim of the review was to provide information on the different approaches used in fabricating poly(methyl methacrylate)/polyaniline copolymers, especially in the form of a thin film. This review also highlights the recent modifications and improvements that can be made during the synthesis process to enhance the conductivity of the copolymers.

Overall, this Review is an up-to-date and well organized. However, there is no adequate summary and comprehensive future prospects. Therefore, I suggest the authors, if possible, add additional critical remarks in the summary.

The manuscript is well written but contains spelling and typing mistakes (some highlighted below)”, which should be corrected.

Line 2, 17, 28, 31, 662: “Poly(methylmethacrylate)”: should be “Poly(methyl methacrylate)”. The IUPAC rules should be used for polymer names. Monomer name consists of two words. Please correct throughout the text.

Line 169: I suggest renaming the section to “PANi-based Blends and Composite”.

Table 2, Table 3 should be written in the same font as the text. Please correct.

Reviewer 3 Report

POLYMERS

Manuscript ID: polymers-1140327

TITLE: Synthesis and Conductivity Studies of Poly (Methylmethacrylate) (PMMA) By Co-polymerization and Blending with Polyaniline (PANi)

AUTHORS: Helyati Abu Hassan Shaari, Muhammad Mahyiddin Ramli, Mohd Nazim Mohtar*, Norizah Abdul Rahman, Azizan Ahmad

Extensive work has been done on representativeness and updating of references on the important issue of different approaches used in fabricating PMMA / PANi copolymers, which the authors deal with.

The work deserves to be published in Polymers after the authors respond to the comments below:

There are five figures in the manuscript, four of which are from specific references; besides, figure 2 has nothing to do specifically with the PMMA/PANI system.

There are eight tables, of which only three are combinational, while the remaining five come from specific references; the title of table 3 is wrong.

Most of the conclusions could be illustrated in Table(s).

The same applies to eventual open-ended questions and challenges, missing

In the sections 8.1 Molarity of acids, 8.2 Emulsion polymerization method, 8.3 Polymerization temperature and, 9. Polymerization yield, any involvement of PMMA is missing.

Lines 143-145: “Since PANi has poor dis-143 persibility, magnetite nanoparticles (for example, Fe3O4) were usually added to improve 144 the compatibility of PANi with PMMA.” Why does it happen?

Lines 148-149: “the addition of iron nanoparticles created uniform 148 dispersion between the PMMA and PANi” Why?

Lines 219-222: “The characterization of the resulted polymers using the FTIR analysis confirmed the interaction of PANi with the PMMA matrix, as the band that is attributed to the stretching vibration of N-Benzenoid-N and N=Quinoid=N at 1485cm-1 and 1144 cm-1, respectively, was observed [44].” What does it mean that?

Line 246: “polyvinyl monomers”. What does it mean that?

Lines 374-376: “In their work, they reported that the conductivity increases linearly with the doping level, with the maximum conductivity of 1.68 x 10-4 (Ω.m)-1 was measured.” Maximum at which doping level?

Minor corrections, although I do not feel qualified to judge about the English language and style with confidence.

Lines 44-46: … with a chemical formula of C6H5NH2, and it can be chemically or electrochemically polymerized to produce a long chain of polyaniline (Figure 1).

I would prefer “(figure 2)” to appear just after the chemical formula:  … with a chemical formula of C6H5NH2 (Figure 1), and it can be chemically or electrochemically polymerized to produce a long chain of polyaniline.

Line 53: I suggest “as one of the favorable polymers” instead of “as favorable polymers”

Line 66: most probably, “the conductivity of conjugated polymers is relatively low” instead of “the conductivity of conjugated polymers is relatively slow”  

Line 83: I suggest, “In addition, the conductivity of the produced copolymer is depending on several factors” instead of “In addition, the copolymer conductivity produced is depending on several factors”

Line 107: I suggest, “Generally, the main focus is to produce new material that” instead of “Generally, the main focuses are to produce new material that”

Lines 198-201: I suggest, “A similar approach was done by Dimitriev et al. (2015) but this study focused on different 199 concentrations of PANi and determined the effects on the morphology, electronic absorption spectrum and the ability for acid doping [40].” Instead of “A similar approach was done by Dimitriev et al. (2015) but this study focused on different 199 concentrations of PANi and determined the effects on the morphology, electronic absorption spectrum and the ability for acid doping [Dimitriev et al., 2020].” No reference with “Dimitriev et al., 2020” exists in the manuscript.

Line 370: Please write, “conductive” instead of “conducive”.

Line 422: Please write, “Subsequently” instead of “Subseuqently”

Line 444: please write, “present” instead of “presence”

Reviewer 4 Report

Shaari et al. summarized the synthesis of PMMA/PANi by blending or copolymerization and the corresponding conductivity studies. A few suggestions are listed below.

  1. As the major route for obtaining PMMA/PANi-based polymer is blending or copolymerizing, these two methods should be compared in a table regarding the pros and cons.
  2. Similarly in Section 4, what is the comparison between graft and block copolymers in terms of pros and cons specifically for PANi?
  3. Section 9 should be combined with Section 4. 
  4. Section 5 should be combined with Section 3. 
  5. If the authors provide the structure of aniline in Figure 1, the structure of methyl methacrylate is equally important to be listed. 

Round 2

Reviewer 3 Report

The manuscript is now worthy of publication in Polymers.

Just a short comment: Lines 844-846. Instead of "In their work, they reported  that the conductivity increases linearly with the doping level, with the maximum conductivity of 1.68 x 10-4 (Ω.m)-1 was measured at 25 wt% benzoic acid." I would propose "In their work, they reported that the conductivity increases linearly with the doping level, with the maximum conductivity of 1.68 x 10-4 (Ω.m)-1 was measured at the maximum loading on benzoic acid attempted, at 25 wt%."